# Neonatal Intrahepatic Cholestasis Caused by Citrin Deficiency with *SLC25A13* Mutation Presenting Hepatic Steatosis and Prolonged Jaundice. A Rare Case Report

**DOI:** 10.3390/medicina57101032

**Published:** 2021-09-28

**Authors:** Shu-Wei Hu, Wen-Li Lu, I-Ping Chiang, Shu-Fen Wu, Chung-Hsing Wang, An-Chyi Chen

**Affiliations:** 1Department of Pediatrics, Tungs’ Taichung MetroHarbor Hospital, No. 699, Sec. 8, Taiwan Blvd., Wuqi Dist., Taichung City 435403, Taiwan; t13223@ms.sltung.com.tw; 2Department of Pediatric Gastroenterology, China Medical University Children’s Hospital, No. 2, Yude Rd., North Dist., Taichung City 404327, Taiwan; d0344@mail.cmuh.org.tw; 3Department of Medical Genetics and Pediatric Endocrinology, China Medical University Children’s Hospital, No. 2, Yude Rd., North Dist., Taichung City 404327, Taiwan; u98000023@gap.kmu.edu.tw (W.-L.L.); a22340961@yahoo.com.tw (C.-H.W.); 4Division of Oncology, Department of Pathology, China Medical University Hospital No. 2, Yude Rd., North Dist., Taichung City 404327, Taiwan; d6868@mail.cmuh.org.tw; 5School of Medicine, College of Medicine, China Medical University, No. 91, Xueshi Rd., North Dist., Taichung City 404328, Taiwan

**Keywords:** infant, Pediatrics, cholestasis, fatty liver, *SLC25A13*, NICCD, citrullinemia

## Abstract

*Background*: Neonatal intrahepatic cholestasis caused by citrin deficiency (NICCD) is a rare autosomal recessive disease. The incidence of citrin deficiency is estimated between 1/10,000 and 1/20,000 in Taiwan. *Case report*: This report describes a case of a 42 day old female infant who suffered from prolonged jaundice, poor weight gain, and anemia. The initial total/direct bilirubin levels were 8.1/3.11 mg/dL. Liver biopsy was performed at 47 days old. The pathology revealed lobules marked with macrovesicular and microvesicular fatty metamorphosis. The serum amino acid profile showed elevated levels of threonine, methionine, citrulline, and arginine. Newborn screening disclosed normal results, but the genetic study revealed *SLC25A13* mutation 851–854 del and 615 + 5G > A. The genetic study of her parents showed that the father carried the *SLC25A13* mutation 851–854 del and the mother carried the *SLC25A13* mutation 615 + 5G > A. Treatment with ursodeoxycholic acid decreased the bilirubin levels to a normal range at the age of 5 months. *Conclusion*: This report illustrates that hepatic steatosis is a feature of NICCD. For every young infant patient who develops cholestasis, the pediatrician must consider NICCD as a differential diagnosis even if newborn screening shows normal findings.

## 1. Introduction

Citrin, a protein contributing to the urea cycle, can transport aspartate from the mitochondria to the cytosol so that aspartate and citrulline may form argininosuccinate by argininosuccinate synthase. Lack of citrin causes citrullinemia and hyperammonemia [1] (Figure 1). Citrin deficiency is an autosomal recessive disease divided into two different clinical appearances, i.e., adult-onset citrullinemia type II (CTLN2) and neonatal intrahepatic cholestasis caused by citrin deficiency (NICCD).

CTLN2 was first reported by Miyakoshi et al. as a group of adult diseases with neurological symptoms, hyperammonemia, and citrullinemia [2]. The onset is sometimes sudden and can occur at any time from the age of 20 to 40 years. It can result in progressive neurological symptoms such as disorientation, delirium, mental derangement, and sudden attacks of unconsciousness, and even death within a few years of the onset [3]. The term NICCD was first proposed by Saheki et al., who found that a group of neonates have the same gene mutations with CTLN2. However, the symptoms of NICCD were not as severe as those of CTLN2 and generally improved over time without special treatment [4]. Both CTLN2 and NICCD have the *SLC25A13* mutation that is localized on chromosome 7q21.3 and encodes citrin [4,5]. The incidence of citrin deficiency in Taiwan is estimated between 1/10,000 and 1/20,000. *SLC25A13* encompasses 30 mutation forms, of which 851–854 del and IVS6 + 5G > A are the most common types in Chinese populations [6]. A study reported that the mutation 851–854 del accounted for 70% and the mutation IVS6 + 5G > A accounted for 23% of Chinese cases [7].

## 2. Case Presentation

A female infant was born at a gestational age of 41 weeks with a birth body weight of 2630 g. She was born from a 25 year old healthy Taiwanese mother and was the mother’s first child. There were no abnormal findings in the prenatal examination, and no specific family histories were noted. After birth, she was fed well with breast milk and infant formula. However, light yellowish stool and jaundice were detected later. Hence, the infant was admitted at 42 days old due to prolonged jaundice. The physical examination revealed the following vital signs: temperature 36.6 °C, heart rate 158 beats per minute, respiratory rate 38 per minute, blood pressure 88/66 mmHg, fair activity, poor body weight gain (3245 g), pale looking, icteric sclera, no hepatomegaly, and no abnormal neurological signs.

The laboratory data showed complete blood count with differential white blood count 19,900/μL, hemoglobin level 7.8 g/dL, mean cell volume 79.2 fL, platelets 587,000/μL, neutrophils 28.9%, lymphocytes 60.3%, monocytes 4.1%, total bilirubin 8.1 mg/dL, direct bilirubin 3.11 mg/dL, gamma glutamyltransferase (GGT) 232 IU/L, alkaline phosphatase 1222 IU/L, aspartate aminotransferase (AST) 68 IU/L, alanine aminotransferase (ALT) 37 IU/L, blood ammonia 224 µg/dL, albumin 3.2 g/dL, prothrombin time 12.8 s, and activated partial thromboplastin time 33.8 s. We also examined the presence of several pathogens associated with neonatal hepatitis, including varicella-zoster virus, herpes simplex virus, cytomegalovirus, rubella, Epstein–Barr virus, toxoplasma, hepatitis B virus, and hepatitis C virus, but no positive results were found. Ursodeoxycholic acid at 18.5 mg/kg/day was administered for cholestasis. Blood transfusion was performed with leukocyte-reduced red blood cells 9.2 mL/kg twice to treat anemia, and the hemoglobin level was corrected to 13.6 g/dL.

A liver biopsy was performed at 47 days old, and the pathology revealed fatty metamorphosis. The lobules were marked with macrovesicular and microvesicular fatty metamorphosis without spotty necrosis under hematoxylin and eosin stain (Figure 2). No portal fibrosis or bile ductular proliferation was observed by Masson’s trichrome and reticulin stains. Periodic acid–Schiff staining showed a negative result, indicating no abnormal polysaccharide accumulation in the liver (Figure 3). Biliary atresia was unlikely from the finding of liver pathology.

We evaluated the lipid profile for the findings of neonatal steatosis, which revealed the following data: total cholesterol 224 mg/dL, triglyceride 152 mg/dL, high-density lipoprotein 13 mg/dL, and low-density lipoprotein 152 mg/dL. Total galactose was not measured because the newborn screening showed no galactosemia. The serum amino acid profile, including alanine, serine, proline, valine, threonine, cysteine, leucine/isoleucine, methionine, histidine, phenylalanine, tyrosine, aspartate, glutamate, tryptophan, ornithine, glutamine/lysine, citrulline, glycine, and arginine, was also evaluated, which showed elevation in threonine, methionine, citrulline, and arginine levels. Consequently, multi-aminoacidemia was detected. Because the pathological findings showed no biliary atresia profile and the subsequent laboratory data showed no change in bilirubin levels, liver enzyme, or GGT levels, she was discharged under treatment with ursodeoxycholic acid (15–20 mg/kg/day) and was followed up monthly at the outpatient department. The subsequent bilirubin and GGT levels showed gradual improvement at the outpatient department. Abdominal sonography revealed increasing echogenicity of the liver, which was compatible with fatty liver (Figure 4). The genetic study performed 2 months later revealed the *SLC25A13* mutation 851–854 del and 615 + 5G > A, but the initial newborn screening showed normal findings. The newborn screening surveyed 21 diseases, including glucose-6-phosphate dehydrogenase deficiency, congenital hypothyroidism, phenylketonuria, homocystinuria, galactosemia, congenital adrenal hyperplasia, maple syrup urine disease, medium-chain acyl-CoA dehydrogenase deficiency, glutaric aciduria type I, isovaleric acidemia, methylmalonic acidemia, citrullinemia type I, CTLN2, 3-hydroxy-3-methyl-glutaric acidemia, holocarboxylase synthetase deficiency, propionic acidemia, carnitine transporter defect, carnitine palmitoyl transferase deficiencies, carnitine tanslocase deficiency, very-long-chain acyl-CoA dehydroxygenase deficiency, and glutaric acidemia type II. We also conducted a genetic study for her parents and found that the father carried the *SLC25A13* mutation 851–854 del, and the mother carried the *SLC25A13* mutation 615 + 5G > A. Based on the findings of cholestasis, citrullinemia, argininemia, neonatal steatosis, and *SLC25A13* mutation, we diagnosed NICCD. The patient was well and her bilirubin levels decreased to normal range at 5 months old under treatment with ursodeoxycholic acid. We reported this case after obtaining appropriate informed consent from the parents and ensured the privacy and security of the patient.

## 3. Discussion

Newborns suffering from jaundice at >14 days are frequently noted at the pediatric clinic, and this is considered prolonged jaundice. Most often, jaundice will subside before 1 month old; however, if it still persists, it is necessary to check the serum bilirubin level, including total and direct forms, of the patient to differentiate the condition from conjugated hyperbilirubinemia. If the infant has conjugated hyperbilirubinemia, we must evaluate common blood cell counts and the levels of AST, ALT, alkaline phosphatase, GGT, albumin, blood ammonia, prothrombin time, and activated partial thromboplastin time, and perform tests for toxoplasma, herpes simplex virus, rubella, cytomegalovirus, and abdominal sonography. A GGT concentration of >300 IU/L or a GGT/AST ratio of >2 indicates a high possibility of biliary atresia [8,9]. If the abdominal sonography shows triangular cord sign or absent gallbladder, biliary atresia is also highly suspected [10,11]. For infants with highly suspected biliary atresia, magnetic resonance cholangiopancreatography is a better method to confirm biliary atresia. In case magnetic resonance cholangiopancreatography is not available, a liver biopsy may provide sufficient information to differentiate biliary atresia, hepatitis, Alagille syndrome, or other diseases such as NICCD (Figure 5).

Patients with NICCD generally suffer from prolonged jaundice, failure to thrive, hepatomegaly, diffuse fatty liver, parenchymal cellular infiltration associated with hepatic fibrosis, variable liver dysfunction, hypoproteinemia, coagulopathy, hemolytic anemia, elevated alpha-fetoprotein level, hypoglycemia, multi-aminoacidemia (citrulline, arginine, threonine, methionine, tyrosine, and elevated threonine/serine ratio), and galactosemia before 1 year old [12].

Yeh et al. analyzed 68 infant patients with idiopathic intrahepatic cholestasis in Taiwan and found that 11 of the 68 infants had steatosis. Among those 11 patients, they performed the *SLC25A13* study for six infants and found that all had the *SLC25A13* mutation. Therefore, they stated that hepatic steatosis can be considered a marker for NICCD in Asian infants with intrahepatic cholestasis [13].

In the present case, the 42 day old female infant was admitted to the children’s hospital due to prolonged jaundice and had conjugated hyperbilirubinemia (total bilirubin level 8.1 mg/dL, direct bilirubin level 3.11 mg/dL). She underwent a series of studies to differentiate the etiologies of conjugated hyperbilirubinemia, such as biliary atresia, Alagille syndrome, storage disease, and metabolism disease. Cholestasis, citrullinemia, argininemia, and neonatal steatosis were detected in this young infant, and the genetic study revealed the *SLC25A13* mutation 851–854 del and 615 + 5G > A. The final diagnosis was NICCD. However, the newborn screening for this patient disclosed a normal result of CTLN2. Therefore, our findings indicate that we must consider the diagnosis of NICCD for patients with a similar clinical picture, even if the newborn screening shows normal results.

Infants with NICCD may require medium-chain triglyceride (MCT) formula and supplementation with lipid-soluble vitamins. MCT can produce energy efficiently even under the cholestatic condition so that it can improve the failure to thrive and cholestasis condition. The absorption of lipid-soluble vitamins, such as vitamins A, D, and E, is poor under cholestatic status, and hence persistent cholestasis may cause the deficiency of lipid-soluble vitamins with related symptoms, such as ocular symptoms, rickets, peripheral neuropathy, and cerebellar ataxia [3].

Several metabolic diseases, such as Wilson’s disease, NICCD, ornithine transcarbamylase deficiency, and carnitine deficiency, manifest steatohepatitis and cirrhosis [14]. Whenever an infant with hepatic steatosis and other NICCD profiles, such as cholestasis and multi-aminoacidemia, is encountered, it is recommended to administer the MCT formula and lipid-soluble vitamin supplement early before the results of genetic study become available as the genetic study requires a significant amount of time.

## 4. Conclusions

Numerous etiologies manifest as conjugated hyperbilirubinemia. Biliary atresia, which is the most important etiology, should be excluded as early as possible. Because valuable time is required for treatment, delayed diagnosis could lead to poor prognosis. Pediatricians must try their best to explore etiologies in their facility for every young infant with cholestasis. Liver biopsy, metabolic profiling, and genetic studies are powerful diagnostic tools that can help differentiate and diagnose the disease. Although the young patient with NICCD had jaundice for several months, her clinical course was good under ursodeoxycholic acid treatment. Therefore, for every young infant with cholestasis, the pediatrician must consider NICCD as a differential diagnosis, even if the newborn screening shows normal findings.

## Figures and Tables

**Figure 1 medicina-57-01032-f001:**
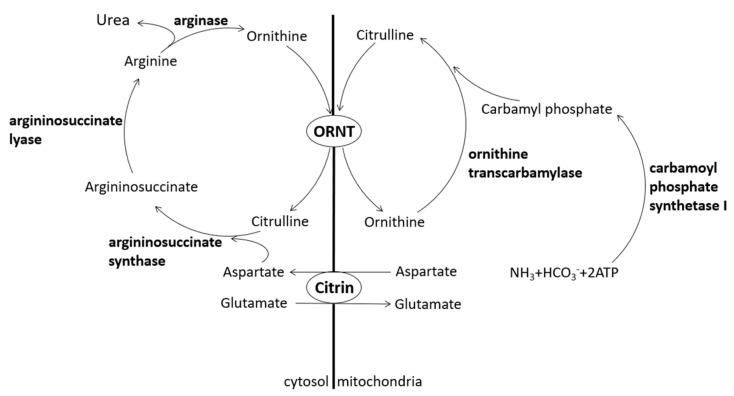
Urea cycle and citrin. Citrin acts as an aspartate–glutamate carrier. ORNT, ornithine/citrulline transporter.

**Figure 2 medicina-57-01032-f002:**
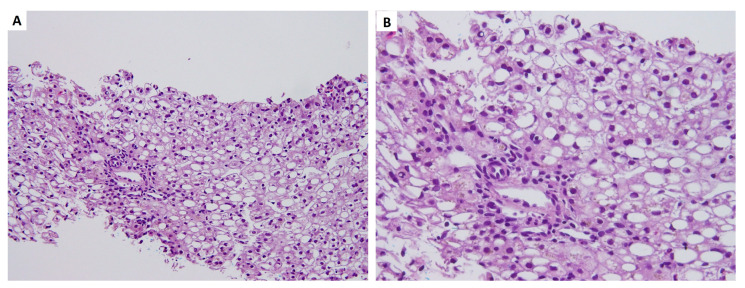
Hematoxylin and eosin staining showed that the lobules marked macrovesicular and microvesicular fatty metamorphosis with no spotty necrosis. (**A**) as observed under 200× and (**B**) as observed under 400×.

**Figure 3 medicina-57-01032-f003:**
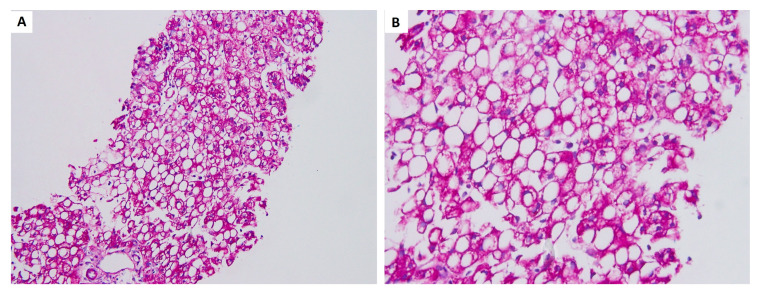
Periodic acid-Schiff staining showed negative result. (**A**) as observed under 200× and (**B**) as observed under 400×.

**Figure 4 medicina-57-01032-f004:**
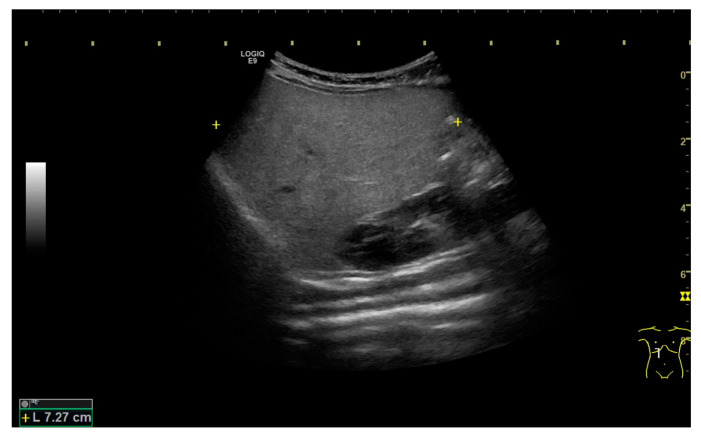
Abdominal sonography showed increasing homogeneous echogenicity of the liver, which is compatible with fatty liver.

**Figure 5 medicina-57-01032-f005:**
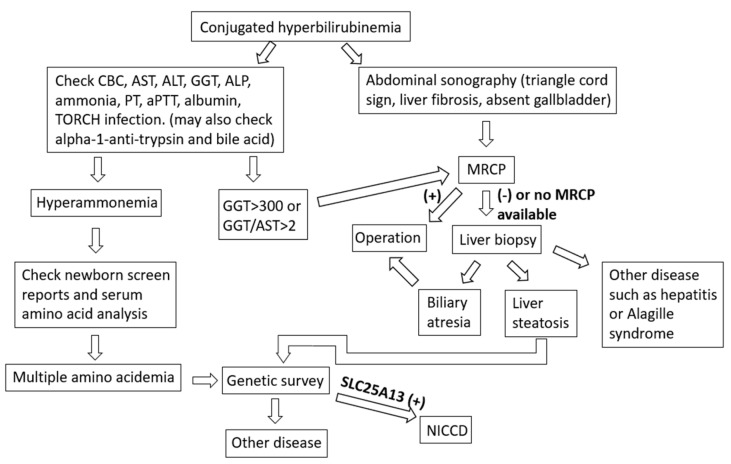
Clinical approach for infants with conjugated hyperbilirubinemia. CBC, common blood cell counts; AST, aspartate aminotransferase; ALT, alanine aminotransferase; GGT, gamma glutamyltransferase; ALP, alkaline phosphatase; PT, prothrombin time; aPTT, activated partial thromboplastin time; TORCH, toxoplasma, others, rubella, cytomegalovirus, herpes simplex virus; MRCP, magnetic resonance cholangiopancreatography; NICCD, neonatal intrahepatic cholestasis caused by citrin deficiency.

## Data Availability

The data that support the findings of this study are available on request from the corresponding author A.-C.C. The data are not publicly available due to their containing information that could compromise the privacy of research participants.

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
