# Peer review of "Neonatal Intrahepatic Cholestasis Caused by Citrin Deficiency with SLC25A13 Mutation Presenting Hepatic Steatosis and Prolonged Jaundice. A Rare Case Report"

_medicina, 2021, doi:10.3390/medicina57101032_

Round 1

Reviewer 1 Report

The authors report a rare case of citrin deficiency presented as neonatal cholestasis. The manuscript has important issues that the authors should resolve.

The Introduction should include data regarding the case's background, presenting the disease, and the possible difficulties in diagnosis, treatment, management. The authors start directly with the case report. All the presentation of the case should be improved. Also, the Discussion section should include more data. The conclusions also should be reviewed and improved. The authors should emphasize why this case should be presented to readers and what is the new idea that this paper brings to the already published literature. 

The use of the English language should be corrected as there are many errors. Also, there are some editing errors. In keywords, there should not be abbreviated words.

Reviewer 2 Report

The authors describe an interresting presentaion of a neonate with Citrin deficiency. Although the case itself is presented in detail, with high quality pictures of stained liver biopsies, and liver ultra sound, I have several major constraints:

  1. The organization of the manuscripts needs a complete revision. The authors start with the description of the case in the Introduction.
  2. In contrast, the general introduction of Citrin deficiency, NICCD, and CTLN2, is found in the discussion (line 123-164)

And some minor:

3. Title: A Rare Case Report and Literature Review.

a) Probably the autors mean: <the clinical presentation with the described symptoms is rare.

b) For a literature review, 8 citations seem quite a small number, with the lates from 2019. In PubMed there are 37 articles searches just for Citrin defiency in 2020-2021.

The author should either delete <Literature Review>, or do a proper literature search and review the relevant articles in the Introduction.

4. The wording <negative NBS> should be avoided, rather <a normal NBS> should be used. In addition the authors must state explicitly what was determined in NBS. At least the results of amino acids and total galactose have to be stated. Or declared that (perhaps) total galactose was not measured.

5. Units should be used properly, preferably SI units. For example line 55 "g" for gramme, not "gm"

6. All abreviations should be explained, when first used, like NICCD (line 19), and not the other way round like MCT in line 153.

7. The meaning of [I] and [Χ] should also be explained when first used. And it should be ommitted, where it has no meaning, like in lines 138-141.

8. The SLC25A13 gene sure has not only 12 mutations. What the authors probably mean is that in literature [5] the frequency and distribution of these 12 mutations are described.

9. In addition to point 8, the authors should at least also cite the paper of Ayako Tabata et al. in J Hum Genet 2008;53:534-545, where another 13 novel mutations are described.

Reviewer 3 Report

The article describes an interesting case report of a newborn suffering from intrahepatic cholestasis caused by citrin deficiency. The article contains a relevant information for readers of medicina.

The English of the article contains many language errors and the case report should be presented in past tense. I recommend to ask for the support of a professional medical writing service to revise the article.

I think a figure showing the biochemical effect of this genetic disorder on citrin metabolism would be informative for readers.

I think for pediatricians and neonatologists it would be fine to have a table or flow chart how to diagnose a newborn with citrin deficiency and biliary atresia. Please highlight the main differences of these diseases.

Line 168: replace “pool” by “poor”

Line 169: replace “your” by “their”

Line 173: replace “young infant patient” by “young infant”

Round 2

Reviewer 1 Report

The paper was improved by the author's changes in all aspects from the recommendations: introduction, case presentation, discussion, corrections of the English language. The manuscript is better now and easy to follow and understand. I have no other recommendations to make.

Reviewer 2 Report

Dear Authors,

thank you for the thorough revision of the manuscript.